# The Impact of Healthy Lifestyles on Late Sequelae in Classical Hodgkin Lymphoma and Diffuse Large B-Cell Lymphoma Survivors. A Systematic Review by the Fondazione Italiana Linfomi

**DOI:** 10.3390/cancers13133135

**Published:** 2021-06-23

**Authors:** Carla Minoia, Chiara Gerardi, Eleonora Allocati, Antonella Daniele, Vitaliana De Sanctis, Alessia Bari, Attilio Guarini

**Affiliations:** 1Hematology Unit, IRCCS Istituto Tumori “Giovanni Paolo II”, 70124 Bari, Italy; attilioguarini@oncologico.bari.it; 2Istituto di Ricerche Farmacologiche “Mario Negri” IRCCS, 20156 Milano, Italy; chiara.gerardi@marionegri.it (C.G.); eleonora.allocati@marionegri.it (E.A.); 3Experimental Oncology and Biobank Management Unit, IRCCS Istituto Tumori “Giovanni Paolo II”, 70124 Bari, Italy; antonella.daniele@oncologico.bari.it; 4Department of Medicine and Surgery and Translational Medicine, “Sapienza” University of Rome, Radio-Therapy Oncology, Sant′ Andrea Hospital, 00189 Rome, Italy; vitaliana.desanctis@uniroma1.it; 5Dipartimento di Scienze Mediche e Chirurgiche Materno-Infantili e dell’Adulto, Università di Modena e Reggio Emilia, 41124 Modena, Italy; alessia.bari@unimore.it

**Keywords:** survivors, classical Hodgkin lymphoma, diffuse large B-cell lymphoma, healthy lifestyle, physical exercise, survivorship care plan, quality of life, systematic review

## Abstract

**Simple Summary:**

With the presented study, Fondazione Italiana Linfomi (FIL) researchers want to fill a gap in the literature regarding long-lived lymphoma patients (beyond 5 years after diagnosis). These patients can develop a series of late sequelae that affect their quality of life and overall survival, especially cardiotoxicity and secondary malignancies. In this context, although part of the risk is closely related to the chemotherapy and radiotherapy, some risk factors can be modified through tertiary prevention. There are currently no specific indications for tertiary prevention in the subset of long-term lymphoma survivors. This systematic review conducted by the FIL researchers is aimed at understanding whether there is evidence that correcting unhealthy lifestyles can reduce the onset of late sequelae.

**Abstract:**

Background: In recent years, the scientific community has been paying ever more attention to the promotion of lifestyles aimed at the prevention of late toxicities related to anti-cancer treatments. Methods: Fondazione Italiana Linfomi (FIL) researchers conducted a systematic review in order to evaluate the evidence in favor of the promotion of lifestyles aimed at the prevention of the main sequelae of long-term classical Hodgkin lymphoma (cHL) and diffuse large B-cell lymphoma (DLBCL) in survivors treated at adulthood with first-line or second-line therapy, including autologous stem cell transplants (ASCTs). Pubmed, Embase and Cochrane Library were searched up to December 2020. Results: Seven studies were ultimately included in this systematic review; some of them were eligible for multiple PICOS. The majority of the studies emerged from data extraction regarding cHL; less evidence resulted for DLBCL survivors. Five studies in favor of physical activity provided consistent data for a reduction of the cardiovascular risk in cHL and also in survivors who underwent ASCT. A beneficial effect of physical activity in reducing chronic fatigue was found. Being overweight was associated with a higher risk of coronary heart disease in cHL survivors in one of the two eligible studies. Studies aiming to evaluate the impact of the Mediterranean diet on late toxicities and secondary cancers were lacking. Tailored survivorship care plans (SCP) seemed to represent an optimal tool to guide the follow-up and promote healthier lifestyles in the one eligible study. Thus, promotion of healthy lifestyles and empowering of lymphoma survivors should be implemented through structured models. The study also brought to light numerous areas of future clinical research.

## 1. Introduction

Survivorship is becoming an important issue of cancer care. It is estimated that the number of cancer survivors has now reached more than 16 million in U.S.A. and 12 million in Europe, and has shown a constant increase over time [1,2,3,4,5]. The majority of survivors have been cured for breast, prostate or colorectal cancer, or melanoma—accounting for about the 58% of survivors [2]. In the context of hematological neoplasms, lymphomas represent the population with the greatest survival rate (free from therapy), thanks to the constant amelioration of diagnostic and therapeutic strategies [3,4,5,6,7].

Classical Hodgkin lymphoma (cHL) usually manifests in the second or third decade of life, with an incidence of 2.3–2.6 new cases/100,000/year [8,9]. The disease in now curable in at least 80% of patients, with a 5-year overall survival (OS) of 88% [8,9,10]. It represents one of the cancers for which a remarkable improvement has been observed in the last 40 years [10]. Long-term survivors of cHL have been generally treated with conventional chemotherapy and radiotherapy, and data on the long-term toxicity of new drugs will be known in a short time [8,10]. Diffuse large B-cell lymphoma (DLBCL) is the most common lymphoid neoplasm in adults, and accounts for about the 40% of all non-Hodgkin lymphoma (NHL). Its incidence ranges from 3.8 to 5.6/100,000/year, with a median age at diagnosis of 65–70 years [9]. The 5-year OS is actually 60–64% [3,4,5,6,7,8,9,10]. Survivors of DLBCL have been treated in the majority of cases with standard induction chemotherapy based on a CHOP (cyclophosphamide, vincristine, doxorubicine, prednisone) or rituximab-CHOP regimen according to the historical period. Salvage chemotherapy and consolidation with autologous hematopoietic stem cell transplant (ASCT) represent the second-line treatments for eligible patients with cHL or DLBCL [11,12]. These two cohorts then constitute the prevalent populations of long-term lymphoma survivors [3] and thus the population to which the systematic review we present has been addressed.

Long-term lymphoma survivors could develop a series of late sequelae, mainly represented by cardiovascular, endocrine–metabolic and neurological toxicities; secondary cancers; and infertility [13,14]. The genesis of these late toxicities is multi-factorial and could be related to two main groups of risk factors: (i) non-modifiable factors: chemotherapy, radiation therapy and ASCT, family history and age [15,16]; and (ii) modifiable factors, such as unhealthy lifestyle factors [17,18].

Lifestyles represent an emerging topic both in cancer prevention and in the prevention of late sequelae after chemotherapy and radiotherapy. In this context, it is essential to identify the unhealthy lifestyles of long-term survivors in order to ameliorate educational and preventive measures, reduce the risk of subsequent diseases and improve quality of life (QoL). Most of the published experiences in this sense concern patients who had solid tumors—in particular, breast and colorectal cancers. For some cancer types, a healthy lifestyle has been associated with reduced risks of recurrence and mortality [2]. In the subset of cancer survivorship, individual lifestyle behaviors, such as maintaining normal bodyweight, physical activity, smoking and diet quality, have been associated with reduced mortality [19,20]. The meta-analysis by Mishra et al. demonstrated a beneficial effect of exercise on health-related quality of life (HRQoL) in 40 trials, including heterogeneous cohorts of cancer survivors [21].

Currently, there are not structured indications either for the monitoring of late sequelae in lymphoma survivors or for their prevention by adopting healthy lifestyles, but onco-hematologists refer to cancer survivors’ guidelines in general [2,3,4,5,6,7,8,9,10,11,12,13,14,15,16,17,18,19,20,21,22]. According to these guidelines, it is advised that cancer survivors adhere to regular physical activity (at least 150 min of moderate or 75 min of intense physical activity per week), to a Mediterranean diet, control bodyweight and abolish smoking [2,3,4,5,6,7,8,9,10,11,12,13,14,15,16,17,18,19,20,21,22,23,24]. Several studies suggest a high prevalence of unhealthy lifestyle factors among lymphoma survivors and the positive effects of healthy lifestyles on different clinical outcomes. For example, about two thirds of patients seem not to perform regular physical activity (75.3%) [18,25] or follow the Mediterranean diet (89%); one third are overweight (BMI 25–29.9, 30.8%) or obese (BMI > 30, 26.4%); and 15.7% keep up smoking habits [9]. In several patients (64.7%), multiple unhealthy lifestyle factors are present [18,26]. Similar features were also reported in a large cohort of long-term lymphoma survivors after ASCT and were associated with more comorbidities [27]. Some unhealthy lifestyle factors seem to affect different aspects of the patient’s life. It has been reported that regular physical activity ameliorates QoL and improves psychological and physical fitness in cHL and NHL survivors [25,26,28]. Moreover, a large population study conducted on 3970 B-NHL showed that normal bodyweight is associated with a reduced risk of developing diabetes mellitus [29]. Obesity represented a risk factor for cardiovascular diseases in a cohort of 2617 5-year cHL survivors [30]. Survivors of NHL who met the American Cancer Society’s (ACS) health-related guidelines appeared to have better QoL and less incidence of post-traumatic stress disorder [24,26]. Chronic fatigue has been documented in about 30% of cHL survivors and is associated with early cognitive impairment [31,32]. In some cases, it has been shown that the modification of lifestyles, i.e., the introduction of regular physical activity, may reduce many of these late events [33,34].

In the last few years, survivorship care plans (SCP) have been developed as an instrument able to guide the follow-up of cancer survivors in a more homogeneous and standardized way [35]. Each SCP includes a summary of the oncologic history and comorbidities; the indications for detecting late recurrences of the hematologic disease and monitoring of long-term toxicities; and the recommendations for secondary cancer screening and vaccinations. SCP should also contain tips for a healthy lifestyle. Despite the existence of electronic systems for the generation of SCP, these tools are not yet routinely applied during the follow-up of lymphoma survivors [15]. According to Majhail et al., the application of SCP to a population of patients after autologous or allogeneic stem cell transplant ameliorated QoL and reduced distress levels [36]. The use of comprehensive SCP could determine greater compliance with screening exams for secondary cancers and late toxicities, and greater adherence to healthy lifestyles, thereby increasing patients’ empowerment. An advanced model of nationwide SCP is represented by the Dutch BETER consortium, which is ongoing and has the aim of educating the population of cHL survivors about the late effects of treatments [37].

With these premises, Fondazione Italiana Linfomi (FIL) researchers aimed to evaluate, through a systematic review of the literature, possible correlations between lifestyles and late sequelae of chemotherapy and radiotherapy in a population of long-term cHL and DLBCL survivors. The reasons for the inclusion of these two populations in the systematic review, which have been extensively discussed within the multi-disciplinary working group, resided in the epidemiology, the cure rates and the possibility of analyzing cohorts of homogeneous and homogeneously treated patients, for whom long-term toxicities are more evaluable. The following late toxicities were considered in the search: cardiovascular, endocrine–metabolic (diabetes, metabolic syndrome/sarcopenia/osteoporosis), neurological (chronic fatigue/neuropathy/cognitive impairment) and secondary cancers. Possible correlations with QoL and OS have also been sought. The systematic review also included a search question on the effectiveness of SCP administration, in terms of reduction of late toxicities, amelioration of compliance with screening programs and improvement of QoL.

## 2. Materials and Methods

This systematic review is part of a series of analyses exploring the management and follow-up of long-term lymphoma survivors and supporting the FIL position statements. The scope of the position statements, the clinical questions and the population/intervention/control/outcome/study design (PICOS) for each question were discussed and agreed on by the FIL “Long-term Survivors Committee” and presented at the FIL congress in 2019 (details on clinical questions and PICOS are reported in Table 1. We used the Preferred Reporting Items for Systematic reviews and Meta-analyses (PRISMA) guidelines to report the results [38]. FIL researchers have been supported by a methods team, including three expert researchers in clinical research methodology and systematic reviews (Centre for Health Regulatory Istituto Di Ricerche Farmacologiche “Mario Negri”, Milan).

### 2.1. Study Identification

MEDLINE (via PubMed), the Cochrane Library and EMBASE were systematically searched from January 1990 to December 2020, with no language or publication type restrictions. Search terms included extensive controlled vocabulary (MeSH and EMTREE); and free-text keywords, combining the conditions (classical Hodgkin lymphoma and diffuse large b-cell lymphoma), interventions (e.g., chemotherapy and radiotherapy) and outcomes of interest (e.g., cardiotoxicity, metabolic syndrome and fatigue). Details on the search strategies can be found in Appendix A. We checked the reference lists of relevant studies to retrieve further studies and congress abstracts and searched study registries for unpublished or ongoing studies.

### 2.2. Eligibility Criteria

We included both primary studies (randomized controlled trials; prospective and retrospective cohort studies; and registry studies) and systematic reviews including these study designs. We included studies involving long-term (≥5 years disease-free or treatment-free) adult (≥18 years-old at diagnosis) cHL or DLBCL survivors. Included studies assessed the impact of modifiable healthy lifestyle factors on the incidence and outcomes of long-term effects on cardiac function, metabolic syndrome/diabetes/sarcopenia/osteoporosis, chronic fatigue/neuropathy/cognitive impairment, secondary cancers, QoL and OS. In parallel, we evaluated the impact of the application of programmed SCP on the above-mentioned outcomes and on the compliance with secondary cancer screenings. The evaluation focused on patients treated with first-line therapy or second-line therapy, including ASCT; allogeneic stem cell transplant was an exclusion criterion.

### 2.3. Study Selection and Data Extraction

One reviewer screened the titles and abstracts to select the studies; reviewed the full-text publications to confirm the eligibility; and extracted the relevant information from the included trials (C.M.). A second reviewer checked the eligibility and the data extraction to increase the accuracy of the process (A.B.). Any discrepancies were resolved by consensus and arbitration by a third author (G.C.). Data collected from each study comprised the following predefined items: (1) study identifier (first author, year of publication); (2) reference; (3) other publication; (4) study design; (5) population;(6) study duration; (7) follow-up; (8) sample size; (9) intervention/control group; (10) outcome measure; (11) main results; (12) conclusion; (13) risk of bias/quality assessment. A predefined spreadsheet (Excel 2007, Microsoft Corporation^®^) was used for data extraction. Data selection and extraction are presented in Appendix A.

### 2.4. Risk of Bias and Quality of Evidence Assessment

We assessed the methodological quality of the included systematic reviews using the AMSTAR 2 tool [39]; the risk of bias for the RCTs using the Cochrane Risk of Bias (ROB) [40]; and the quality of cohort, case-control and registry studies using the New Castle Ottawa Scale [41]. The risk of bias and quality of evidence assessment was done by one reviewer and checked by a second (Appendix A).

### 2.5. Data Synthesis

As we expected a substantial degree of heterogeneity among the included studies, we did not pool data in meta-analyses. For each clinical question, the pertinent PICOSweredefined and the included studies providing relevant information were summarized narratively and tabulated to highlight similarities and differences in methods and results. We focused on the review outcomes.

## 3. Results

### 3.1. Physical Activity/Exercise: Does Regular Physical Activity/Exercise (at Least 150 min/week of Moderate Physical Activity or 75 min/week of Intense Physical Activity) Determine a Clinical Benefit in Long-Term cHL or DLBCL Survivors?

The initial search produced 424 potentially relevant articles. One additional record was identified through other sources. After removing duplicates, 401 abstracts were assessed for eligibility and 41 were admitted to subsequent assessment. Of these, five studies were included in the final analysis after full-text evaluation. The main reasons for exclusion were: the presence of a pooled population including different histotypes of NHL; a population not matching the definition of long-term survivorship; the lack of an intervention impacting on outcome. Details of the screening process, including reasons for full-text exclusion, are reported in Figure 1 (PRISMA flow-chart).

Four studies were conducted from 1999 to 2014 [30,31,42,43], and the period of evaluation was not reported for the remaining one [44]. Four studies focused on cHL survivors [30,31,42,44] and one on the population of long-term survivors after ASCT [43]. 

One nested case-control study presented the cardiovascular late toxicity outcome and discussed the impacts of lifestyle factors [30] in a large cohort of 2617 5-year cHL survivors treated between 1965 and 1995. Follow-up was complete up to October, 2013. The study evaluated the risk of coronary heart disease (CHD) as the first cardiovascular event after lymphoma, according to the radiation dose to the heart and type of chemotherapy. An estimate of the in-field irradiated heart volume was calculated via radiation charts and simulation radiographs. Other clinical and lifestyle factors were collected from medical records and from mailed questionnaires completed by the general practitioners (70% of responses). From the whole cohort, 325 survivors reported a CHD as the first event (median interval from lymphoma: 19 years). For each patient with CHD, four controls who had not developed cardiac disease and were matched for sex, age and date of HL diagnosis, had been selected (*n* = 1204). Treatments were variable but controlled for both case and control with multivariate regression analysis. In response to our PICOS, the authors reported that a higher level of physical activity at the time of the follow-up questionnaire was associated with decreased CHD risk (RR 0.52; 95% CI, 0.32 to 0.83). Patients who performed physical activity (>3 h/week of walking, cycling or sports) had considerably lower risks of CHD than inactive patients (<1 h/week). The authors concluded that early management of cardiovascular risk factors and encouragement of physical activity might reduce CHD in cHL survivors [30]. The overall quality of the study, according to the Newcastle–Ottawa Scale (NOS), was high, due to an optimal case definition and representativeness; good selection of controls; comparability and ascertainment of exposure of cases and controls; and the method of evaluation of the exposure.

The remaining three studies conducted in the cHL survivor population evaluated the correlation between physical activity and chronic fatigue [31,42,44]. In the cross-sectional study by Ng et al., lower physical activity frequency was found to be associated with increased chronic fatigue in 511 cHL survivors (median 15 years from diagnosis, median age at evaluation 44). Patients were treated from 1969 to 1996. The levels of fatigue were measured by the Functional Assessment of Chronic Illness Therapy-Fatigue (FACIT-F) questionnaire (60.6% of responses). Cardiac disease, tobacco use and psychiatric condition were also associated with referred chronic fatigue. The control population was represented by patients’ siblings, a non-exposed population [42]. The quality according to the NOS of this study was in general intermediate. The case definition was good; however, the selection and exposure of controls were not optimal; the main bias was represented by the self-reported exposure.

A cohort study involved 476 cHL survivors treated from 1971 to 1997, evaluated for the presence of chronic fatigue during the follow-up through the Fatigue Questionnaire (FQ). The frequency of physical activity did not differ among cHL patients experiencing fatigue (30%) compared to cHL not experiencing fatigue. In comparison with the general population (56,999 Norwegian people), male cHL survivors reported increased levels of physical activity, thereby probably contributing to the negative association [31]. A pilot cohort study confirmed the efficacy of physical activity in reducing chronic fatigue (measured by FQ) in a small population of cHL survivors. The study involved nine fatigued subjects enrolled in a home-based exercise intervention lasting 20 weeks; no controls were considered in the analysis [44]. According to the NOS, the first study was intermediate in quality, and the last was low in quality.

The last cross-sectional study was conducted in a population of long-term survivors after ASCT [43]. This study included 194 long-term survivors treated from 1987 to 2008 (mean follow-up 10.2 months) and who were not affected by heart failure. They underwent cardio-respiratory fitness measurements, which were reported in peak oxygen consumption (VO_2peak_) and correlated with medical records and lifestyle factors. The VO_2peak_ in the study population was globally lower than in the general matched population. A direct correlation with physical exercise was found: Inactive survivors presented a significantly lower percent predicted (PP) VO_2peak_ (females: PP 87.5, 95% CI 78.8–96.3; males: PP 85.8, 95% CI 82.0–89.7) than the general population. Values of patients performing regular physical exercise did not differ from those of the reference population (females: PP 105.6, 95% CI 94.8–116.5; males: PP 100.9, 95% CI 94.0–107.8). This benefit was also confirmed for patients who had received doxorubicin, whose late effect was mitigated in moderately and highly active people [43]. It should be noted that the study lacked detailed reporting on histotypes of NHL, and therefore did not allow us to assess which outcomes referred to DLBCL. However, the study met the inclusion criteria, was unique in the subset of ASCT and provided remarkable conclusions in reference to specific chemotherapeutic agents, e.g., doxorubicin. The quality of this study (NOS) was globally high: the representativeness, selection and exposure were good, as were the measurements of the outcome and the duration of the follow-up.

### 3.2. Mediterranean Diet/Nutritional Intervention: Does a Controlled Diet (Mediterranean Diet or Nutritional Plan/Intervention) Determine a Clinical Benefit in Long-Term cHL or DLBCL Survivors?

We screened 88 abstracts, and then four relevant publications were retrieved as full text. All these publications were excluded from the final analysis because the populations did not correspond perfectly to the PICOS (*n* = 3) or the outcome was not evaluable (*n* = 1). Details of the whole screening process, including reasons for full-text exclusion, are reported in Figure 2 (PRISMA flow-chart).

Thus, we did not retrieve studies assessing a controlled diet’s effects on the outcomes defined by the PICOS in long-term cHL or DLBCL survivors.

### 3.3. Bodyweight and BMI: Does a Controlled Bodyweight or Adequate BMI Determine a Clinical Benefit in Long-Term cHL or DLBCL Survivors?

We screened 155 abstracts. One additional record was identified through other sources. Fourteen relevant publications were then retrieved as full text. Of these, 12 were excluded for these main reasons: the population did not perfectly meet the PICOS (*n* = 9); the outcome was not evaluable or correspondent with the PICOS (*n* = 1); the intervention was missing (*n* = 2). Two studies were included in the final sample and relative analysis. Details of the whole screening process, including reasons for full-text exclusion, are reported in Figure 3 (PRISMA flow-chart).

The nested case-control study by van Nimwegen et coll. (previously described) conducted in a cohort of 2617 5-year cHL survivors reported that obesity represented a risk factor for CHD (RR 1.64; CI, 1.24 to 2.16) at the time of the follow-up questionnaire [30].

The second study was a retrospective cohort study conducted in 2009–2010 [45]. Data have been presented only in an abstract. The main outcome of this analysis was to quantify the associations between known cardiac risk factors and the incidence of cardiac disease in a population of 6658 cHL patients treated with anthracycline-containing chemotherapy with or without radiotherapy within nine randomized trials of the EORTC (European Organization for Research and Treatment of Cancer) and GELA (Group d’Etude de Lymphomes de l’Adult, now LYSA) (1964–2004). Of the whole population, 1990 patients responded to a patient-reported questionnaire (median age at treatment 29 years; median follow-up duration 14 years). The analysis showed that BMI ≥ 30 was not statistically significant as a risk factor for cardiac disease (OR 1.3 95%CI 0.9–1.8) [45]. The quality of this study was intermediate: selection of cases and outcome were good; the main concern/cons regarded the lack of a control group and the self-reported outcome. It must be reiterated that the results have been reported only through an abstract, whose full data have not yet been published.

### 3.4. Dietary Supplements: Does the Use of Dietary Supplements Determine a Clinical Benefit in Long-Term cHL or DLBCL Survivors?

We screened 26 abstracts and then two relevant publications were retrieved as full text. All these publications were excluded from the final analysis because the populations did not correspond perfectly with the PICOS (*n* = 2). Details of the whole screening process, including reasons for full-text exclusion, are reported in Figure 4 (PRISMA flow-chart).

### 3.5. Chronic Fatigue/Cognitive Impairment: Does the Use of Non-Pharmacological Preventive Interventions (Physical and Mental Exercise) Reduce the Incidence of Chronic Fatigue and Cognitive Impairment in Long-Term cHL or DLBCL Survivors?

We screened 122 abstracts. Four additional records were identified through other sources. Fifteen relevant publications were then retrieved as full text. Of these, 12 were excluded and three studies were included in the final sample and relative analysis. The main reasons for exclusion were: population including not only cHL or DLBCL or long-term survivors; and being patients treated at pediatric age (*n* = 9); missing interventions (*n* = 2); and being an ongoing study without available results (*n* = 1). Details of the whole screening process, including reasons for full-text exclusion, are reported in Figure 5 (PRISMA flow-chart).

According to this specific PICOs, we did not find studies in response to possible lifestyle interventions preventing cognitive impairment in long-term cHL and DLBCL survivors.

Data on chronic fatigue have partially been reported in Section 3.1. No other lifestyle factors than physical exercise seem to have a beneficial role in the prevention of chronic fatigue in the selected population. As already exposed, the largest studies presented contrasting results: the first showed a beneficial role [42] and the second one highlighted no role [31] of physical activity in chronic fatigue in long-term cHL survivors. The quality (NOS) of these studies was in general intermediate. Moreover, a pilot cohort study showed the efficacy of physical activity in reducing chronic fatigue in a small population of cHL survivors [44]; the study was of globally low quality, as explained before.

No studies specifically addressed the issue of preventive measures for chronic fatigue among DLBCL survivors.

### 3.6. Survivorship Care Plans (SCP): Does the Use of SCP Determine a Clinical Benefit in Long-Term cHL or DLBCL Survivors?

The initial search produced 88 potentially relevant articles. After removing duplicates, we assessed 81 abstracts for eligibility, and only eight full texts were admitted for subsequent assessment. One study was included in the final sample after full text evaluation. The main reasons for the exclusion were: the population did not correspond perfectly with the PICOS (*n* = 3); the outcome was not evaluable (*n* = 3); the intervention was missing or not specified (*n* = 1). Details of the whole screening process, including reasons for full-text exclusion, are reported in Figure 6 (PRISMA flow-chart).

The included study was a prospective cohort study, published as an abstract, with relevant limitations. The main outcome of the analysis was to enhance the awareness and adoption of healthy lifestyles in a population of cHL survivors through a nurse-led survivorship intervention, including the development and delivery of a tailored SCP, and an evaluation by the General Health Index and Health Promoting Lifestyle Profile II at four time points. The nurse-led consultations also included an education package tailored to individuals’ health needs. This study was published in 2013 and recruited 30 cHL survivors and 30 healthy controls. In response to the defined PICOS, from baseline to six months post-intervention, statistically significant benefits were detected for some domains: improvement of physical activity (*p* = 0.014); nutrition (*p* = 0.0005); health promoting lifestyle (*p* = 0.005). The intervention was feasible and demonstrated significant potential to improve awareness of health status and healthy lifestyle behaviors in this subset of patients [46]. It cannot be concluded that the benefits presented by the patients were due only to the administration of the SCP, but that the consultations and the educational package may have played a synchronous role. The quality of this study was generally low due to the unclear process of selection of cases and the definition of controls. Results have been partially reported and no full text is currently available.

No other study specifically tailored to the beneficial role of SCP in the population of cHL and DLBCL long-term survivors was found through the revision process.

## 4. Discussion

This systematic review allowed us to evaluate the efficacy of the adherence to healthy lifestyles among the population of cHL and DLBCL survivors. It highlighted modifiable lifestyle factors which through appropriate counseling could lead to the education and empowerment of the patient. These modifiable factors are represented by physical activity according to guidelines (at least 150 min of moderate or 75 intense physical activity per week), a Mediterranean diet or balanced nutrition and appropriate bodyweight (BMI < 25) [2,23,24,47,48]. The aim was to verify any study designed to assess the impacts of these risk factors on the more frequent long-term toxicities in cHL and DLBCL survivors (cardiovascular toxicity, secondary cancer, metabolic toxicities, chronic fatigue and cognitive impairment) and general outcomes (QoL and OS).

The systematic review showed that the most studied population consisted of cHL survivors, and to a lesser extent of survivors after ASCT. The only lifestyle factor for which there are consistent data is physical activity. In fact, it correlated with a clinical benefit in terms of reduction of cardiovascular risk. CHL survivors who regularly performed physical exercise presented an inferior risk of developing coronary heart disease [30]. Moreover, lymphoma survivors who performed physical activity after ASCT showed an amelioration of their cardiovascular fitness [43]. Physical activity also proved to determine a beneficial effect in reducing chronic fatigue in patients treated for cHL [42,44]. We did not find any data on the correlations among physical activity and other clinical outcomes, e.g., QoL and OS.

Concerning the population of DLBCL survivors, who are more advanced in age, no evaluation supporting improvements via correction of lifestyle on cardiovascular and metabolic outcomes or QoL was available. The major concern was that the analyses performed on DLBCL survivors were often present in larger studies including several NHL histotypes, for which data were not assessed separately, thereby not permitting reliable conclusions for the specific histotype. This was the main limitation in the extraction of conclusive data for DLBCL. Furthermore, the ASCT population was studied very little in this context, even if it could present numerous areas for clinical improvement [36].

Among other factors, a BMI > 25 seemed to be associated with a higher risk of coronary heart disease in cHL survivors [30], even if data were not confirmed in a subsequent study [45]. Both studies correlated BMI > 25 with higher risks for the following: hypertension; diabetes mellitus and smoking during the 5 years preceding the diagnosis of coronary heart disease; and factors related to the oncologic therapy, especially a dose-dependent mediastinal irradiation [30,45]. Probably, risks due to oncologic therapies and other cardiovascular factors overcome the obesity factor in the onset of coronary heart disease in cHL survivors treated before the introduction of modern radiotherapic techniques. However, it is rational to recommend adequate control of bodyweight and if necessary a nutritional intervention, in agreement with the guidelines for survivors of solid tumors and for cardiovascular prevention [2,47,48,49].

Compared to solid tumors, well-designed studies aiming to evaluate the impact of the Mediterranean diet on late toxicities and secondary cancers in long-term cHL and DLBCL survivors are lacking. In any case, general guidelines recommend adhesion to this alimentary regimen for all cancer survivors [2,23,47,48]. Even though from our systematic review no studies had as their topic the benefit of the Mediterranean diet, it is rational to advise a controlled and balanced diet, consistent with the Mediterranean diet, to lymphoma survivors. In this context, there are no evidence to suggest the use of nutritional supplements for the prevention of the defined outcomes in this selected population. During counseling, it would be also important to discuss the abolition of smoking, e.g., suggesting participation in cessation programs, along with the correction of other common cardiovascular risk factors (hypertension, diabetes, etc.), in collaboration with cardiologists.

After the analyses on healthy lifestyles, FIL researchers proposed to evaluate whether there is a rationale for the use of SCP among the population of cHL and DLBCL survivors for ameliorating the diagnosis of late toxicities, adhering to screening programs for secondary cancers and correcting unhealthy lifestyles. SCP are a valid and recapitulatory instrument mainly used in patients healed from solid tumors [35]. Their use could lead to a beneficial role also in lymphoma survivors [15]. Some models of SCP for lymphoma survivors are available online on the principal oncologic scientific society websites [50] and a nationwide program is ongoing for cHL survivors in the Netherlands [37]. Regarding patient empowerment through healthy lifestyles through the use of SCP, there is still little evidence in the literature. A study by Gates et al. showed that the application of SCP via a nurse-led program increased the levels of physical activity among cHL survivors [46]. As previously discussed, precisely this factor led to an amelioration of the cardiovascular toxicity and then of comorbidities. According to the literature, data supporting the benefit of the use of SCP in lymphoma survivors are still insufficient and there is a need for future evaluations.

## 5. Conclusions

In conclusion, the theme of lifestyles in lymphoma survivors is still immature, both in the field of clinical research and in daily practice. Given the clinical benefits of physical activity for cHL survivors, it is desirable to implement programs promoting and educating patients on this topic. In parallel, the areas of development of clinical research are numerous. It would be interesting to investigate modifiable lifestyle factors potentially impacting the prognosis, comorbidities and QoL of DLBCL survivors. Furthermore, a specific topic is survivors after ASCT, for whom clinical research could provide new information about long-term toxicities. For this purpose, tailored SCP according to risk categories could represent the optimal tool. Our analyses focused on patients in remission for more than 5 years. A population similarly significant is represented by patients in remission from 1 to 5 years, a period in which the correction of lifestyles could be more impactful and long-lasting. In this setting, the efficacy of lifestyle correction for cHL and DLBCL survivors would be a promising topic for future research programs.

## Figures and Tables

**Figure 1 cancers-13-03135-f001:**
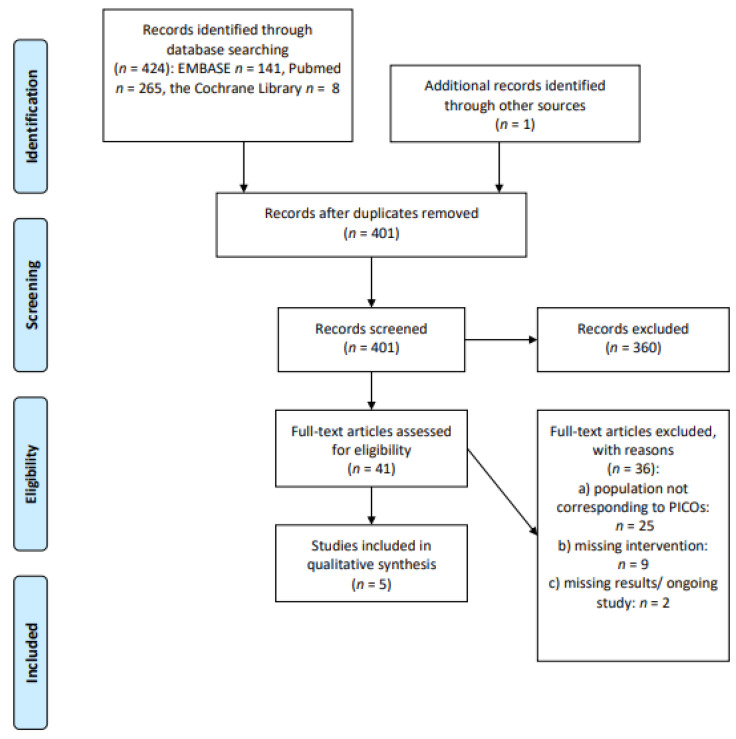
PRISMA flow-chart on physical activity/exercise.

**Figure 2 cancers-13-03135-f002:**
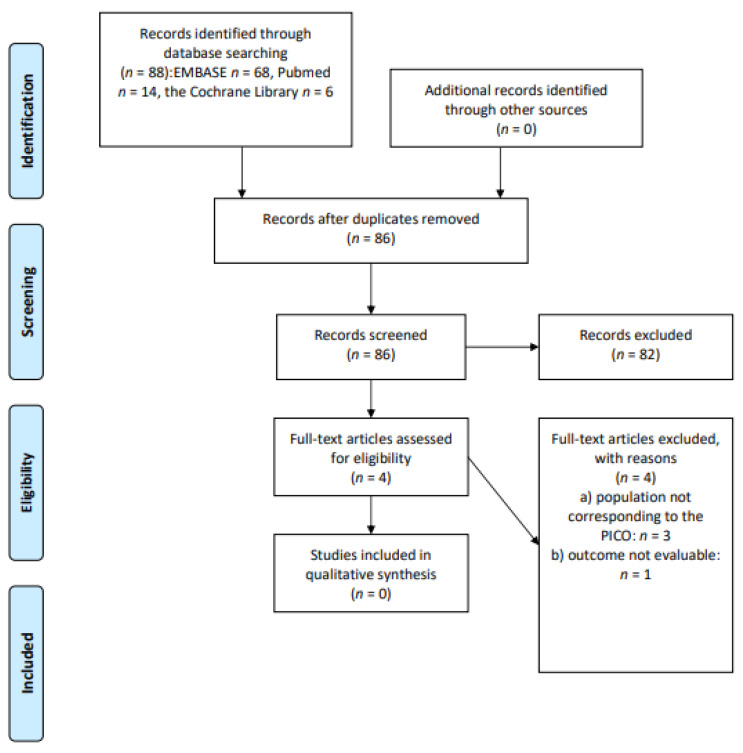
PRISMA flow-chart on Mediterranean diet/nutritional intervention.

**Figure 3 cancers-13-03135-f003:**
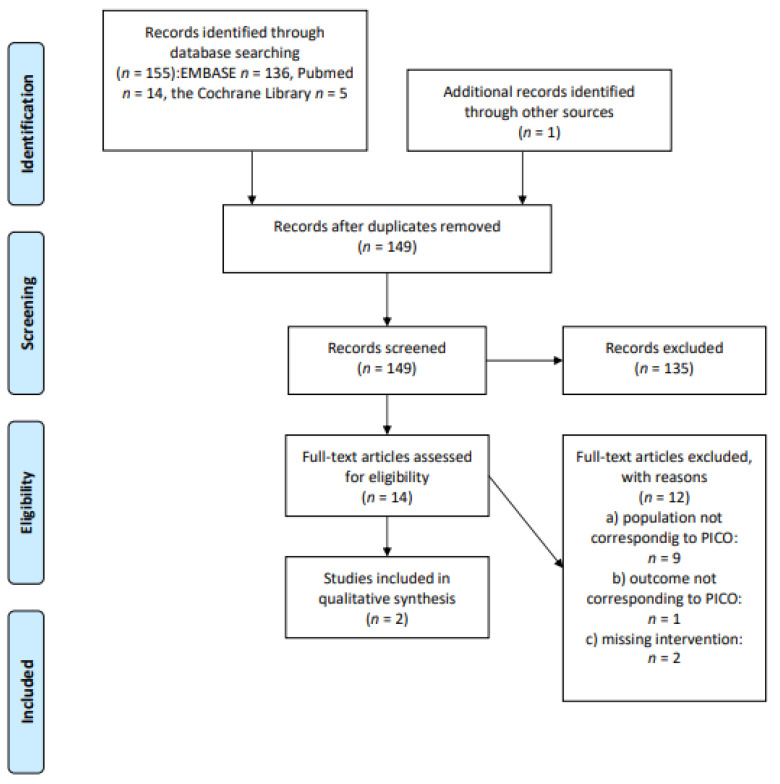
PRISMA flow-chart on bodyweight and BMI.

**Figure 4 cancers-13-03135-f004:**
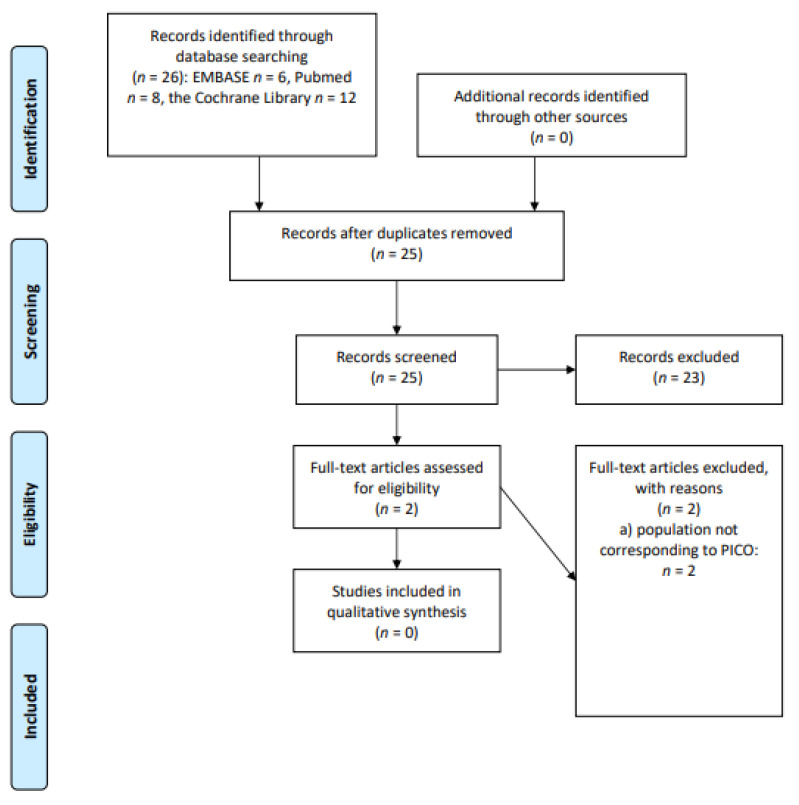
PRISMA flow-chart on dietary supplements.

**Figure 5 cancers-13-03135-f005:**
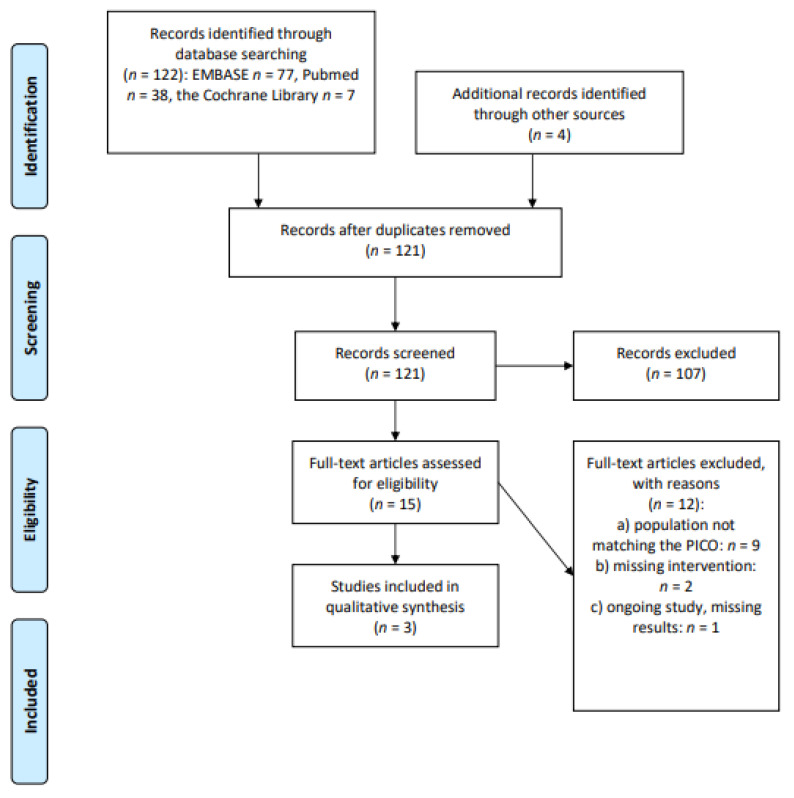
PRISMA flow-chart on preventive measures for chronic fatigue/cognitive impairment.

**Figure 6 cancers-13-03135-f006:**
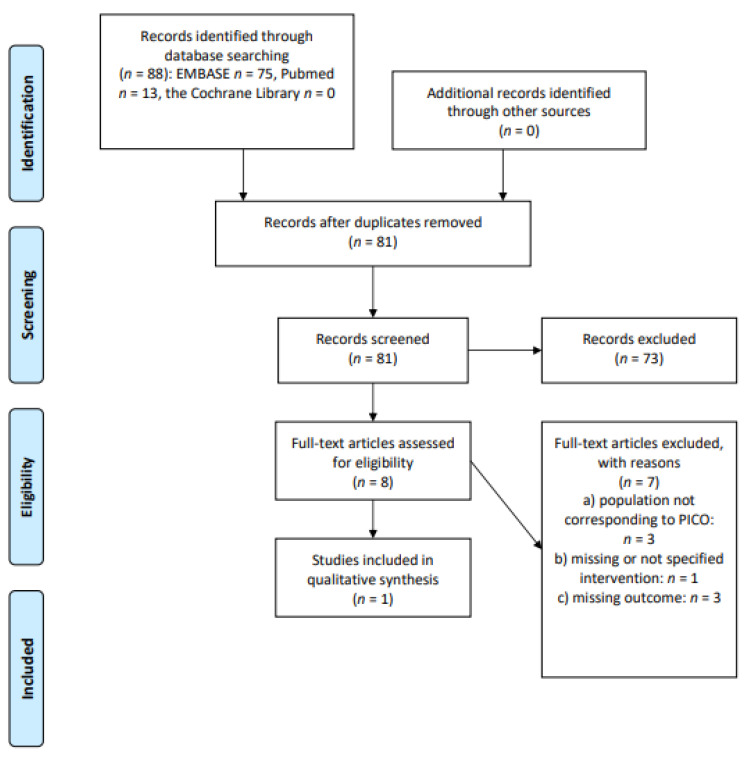
PRISMA flow-chart on survivorship care plans.

**Table 1 cancers-13-03135-t001:** Clinical questions and PICOS addressed by the systematic review: healthy lifestyles and survivorship care plans.

Clinical Question	PICOS
Does a regular physical activity/exercise (at least 150 min/week of moderate physical activity or 75 min/week of intense physical activity) determine clinical benefit in long-term cHL or DLBCL survivors?	*P*: long-term cHL or DLBCL survivors (≥5 years disease- or treatment-free), adults (≥18-year-old at diagnosis) treated with first-line therapy or second-line therapy including ASCT *I*: physical activity/exercise (at least 150 min/week of moderate physical activity or 75 min/week of intense physical activity) *C*: no physical activity/exercise or physical activity *O*: reduced incidence of diabetes/metabolic syndrome/sarcopenia/osteoporosis; reduced incidence of late cardiovascular toxicity; reduced incidence of chronic fatigue/neuropathy/cognitive impairment; reduced incidence of secondary cancers; amelioration of QoL and OS
Does a controlled diet (Mediterranean diet or nutritional plan/intervention) determine a clinical benefit in long-term cHL or DLBCL survivors?	*P*: long-term cHL or DLBCL survivors (≥5 years disease- or treatment-free), adults (≥18-year-old at diagnosis) treated with first-line therapy or second-line therapy including ASCT *I*: controlled diet (Mediterranean diet or nutritional plan/intervention) *C*: no controlled diet *O*: reduced incidence of diabetes/metabolic syndrome/sarcopenia/osteoporosis; reduced incidence of late cardiovascular toxicity; reduced incidence of chronic fatigue, neuropathy, cognitive impairment; reduced incidence of secondary cancers; amelioration of QoL and OS
Does a controlled bodyweight or adequate BMI determine a clinical benefit in long-term cHL or DLBCL survivors?	*P*: long-term cHL or DLBCL survivors (≥5 years disease- or treatment-free), adults (≥18-year-old at diagnosis) treated with first-line therapy or second-line therapy including ASCT *I*: controlled bodyweight or adequate BMI *C*: BMI ≥ 25 *O*: reduced incidence of diabetes/metabolic syndrome/sarcopenia/osteoporosis; reduced incidence of late cardiovascular toxicity; reduced incidence of chronic fatigue, neuropathy, cognitive impairment; reduced incidence of secondary cancers; amelioration of QoL and OS
Does the use of dietary supplements determine a clinical benefit in long-term cHL or DLBCL survivors?	*P*: long-term cHL or DLBCL survivors (≥5 years disease- or treatment-free), adults (≥18-year-old at diagnosis) treated with first-line therapy or second-line therapy including ASCT *I*: use of dietary supplements *C*: no dietary supplements *O*: reduced incidence of diabetes/metabolic syndrome/sarcopenia/osteoporosis; reduced incidence of late cardiovascular toxicity; reduced incidence of chronic fatigue, neuropathy, cognitive impairment; reduced incidence of secondary cancer; amelioration of QoL and OS
Does the use of non-pharmacological preventive interventions (physical and mental exercise) reduce the incidence of chronic fatigue and cognitive impairment in long-term cHL or DLBCL survivors?	*P*: long-term cHL or DLBCL survivors (≥5 years disease- or treatment-free), adults (≥18-year-old at diagnosis) treated with first-line therapy or second-line therapy including ASCT *I*: non-pharmacological preventive interventions (physical and mental exercise) *C*: no non-pharmacological preventive interventions (physical and mental exercise) *O*: reduced incidence of chronic fatigue, cognitive impairment
Does the use of SCP determine a clinical benefit in long-term cHL or DLBCL survivors?	*P*: long-term cHL or DLBCL survivors (≥5 years disease- or treatment-free), adults (≥18-year-old at diagnosis) treated with first-line therapy or second-line therapy including ASCT *I*: use of personalized SCP *C*: no use of SCP *O*: reduced incidence of late toxicities; ameliorated compliance with screening programs for secondary cancers; amelioration of QoL

PICOS, population, intervention, control, outcome, study design; cHL, classical Hodgkin lymphoma; DLBCL, diffuse large B-cell lymphoma; ASCT, autologous stem cell transplant; QoL, quality of life; OS, overall survival; BMI, body mass index; SCP, survivorship care plan.

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
