# Peer review of "The Impact of Healthy Lifestyles on Late Sequelae in Classical Hodgkin Lymphoma and Diffuse Large B-Cell Lymphoma Survivors. A Systematic Review by the Fondazione Italiana Linfomi"

_cancers, 2021, doi:10.3390/cancers13133135_

Round 1

Reviewer 1 Report

Comments – major

  1. The authors of this article intended to focus solely on evidence relevant to survivorship in lymphoma patients; however, given their focus on lifestyle management and survivor care plans, their paper would be strengthened by brief mention of similar research on survivorship in other selected cancers, such as breast cancer, where the impacts of lifestyle interventions and care plans have been examined and, at least in some instances, found to be effective (e.g. Ruiz-Vozmediano, Integr Canc Therap, 2020;19:1-11; Mishra,Cochrane Database Syst Rev. 2012;2012(8):CD008465.)

Comments – minor

  1. The adoption of healthy lifestyles and care plans cannot reduce predisposing factors. They can only reduce the impact of such factors. The authors should consider revising the title to something like “The adoption of healthy lifestyles and Survivorship Care Plans to reduce the impact of factors predisposing to late sequelae in classical Hodgkin lymphoma and Diffuse Large B-cell lymphoma survivors. A systematic review by the Fondazione Italiana Linfomi”
  2. p2, lines 66-67. The word “no” should be removed and the words “nor” and “neither” should be reversed in position to conform to standard English usage.
  3. p11, line 336. ? “relevant” rather than “relative”.

Author Response

Response to Reviewer  1

Comments – major

Question 1

The authors of this article intended to focus solely on evidence relevant to survivorship in lymphoma patients; however, given their focus on lifestyle management and survivor care plans, their paper would be strengthened by brief mention of similar research on survivorship in other selected cancers, such as breast cancer, where the impacts of lifestyle interventions and care plans have been examined and, at least in some instances, found to be effective (e.g. Ruiz-Vozmediano, Integr Canc Therap, 2020;19:1-11; Mishra,Cochrane Database Syst Rev. 2012;2012(8):CD008465.)

Response to question 1

The Authors thank for the suggestion. A reference to solid tumors has been added at line 66 of the introduction:

"Most of the published experiences in this sense concern patients treated from solid tumors, in particular breast and colorectal cancer. For some cancer types, a healthy lifestyle has been associated with a reduced risk of recurrence and mortality [2]. In the subset of cancer survivorship, individual lifestyle behaviors, such as normal body weight, physical activity, smoking, diet quality, have been associated with reduced mortality in (Karav, Ruiz-V). The meta-analysis by Mishra et coll. demonstrated a beneficial effect of exercise on Health Related Quality of Life (HRQoL) in 40 trials including heterogeneous cohorts of cancer survivors (Mishra)."

The following references have been added:

- Karavasiloglou, N., Pestoni, G,; Wanner, M.; Faeh, D.; Rohrmann S. Healthy lifestyle is inversely associated with mortality in cancer survivors: Results from the Third National Health and Nutrition Examination Survey (NHANES III). PLoS One. 2019 26;14(6):e0218048. doi: 10.1371/journal.pone.0218048.

- Ruiz-Vozmediano, J,; Löhnchen, S.; Jurado, L.; Recio, R.; Rodríguez-Carrillo, A.; López, M.; Mustieles, V.; Expósito, M.; Arroyo-Morales, M.; Fernández, M.F. Influence of a Multidisciplinary Program of Diet, Exercise, and Mindfulness on the Quality of Life of Stage IIA-IIB Breast Cancer  Survivors. Integr Cancer Ther. 2020;19:1534735420924757. doi: 10.1177/1534735420924757.

- Mishra, S.; Scherer, R.W.; Geigle, P.M.; Berlanstein, D.R.; Topaloglu, O.; Gotay, C.C.; Snyder, C. Exercise interventions on health-related quality of life for cancer survivors. Cochrane Database Syst Rev. 2012(8):CD007566. doi: 10.1002/14651858.CD007566.pub2.

An introduction to solid tumor has been also written at line 51:

"The majority of survivors have been cured for breast, prostate, colorectal cancer and melanoma, thus accounting for about the 58% of survivors [2].

Comments – minor

Question 2

The adoption of healthy lifestyles and care plans cannot reduce predisposing factors. They can only reduce the impact of such factors. The authors should consider revising the title to something like “The adoption of healthy lifestyles and Survivorship Care Plans to reduce the impact of factors predisposing to late sequelae in classical Hodgkin lymphoma and Diffuse Large B-cell lymphoma survivors. A systematic review by the Fondazione Italiana Linfomi”

Response to question 2

The Authors thank the Reviewer for the suggestion. We made the title more clear:

"The impact of unhealthy lifestyles on late sequelae in classical Hodgkin lymphoma and Diffuse Large B-cell lymphoma survivors. A systematic review by the Fondazione Italiana Linfomi"

Question 3

p2, lines 66-67. The word “no” should be removed and the words “nor” and “neither” should be reversed in position to conform to standard English usage.

Response to question 3

Thank you.

The sentience has been rewritten as follows:

"Currently, there aren't structured indications neither for the monitoring of late sequelae in lymphoma survivors nor for their prevention by adopting healthy lifestyles, but onco-hematologists refer to cancer survivors’ guidelines in general [2-12]."

Question 4

p11, line 336. ? “relevant” rather than “relative”.

Response to question 4

We have modified as follows:

"The included study was a prospective cohort study, published as an abstract with relevant limitations."

Reviewer 2 Report

The authors performed a systematic review and nicely utilized PICOS (patient, intervention, control, outcome, study design) statements to specify the questions being addressed and followed PRISMA guidelines. The content of the review has potential clinical significance for providers of survivorship care and researchers focused on outcomes among adult survivors of adult Hodgkin lymphoma, and less so Diffuse Large B-cell Lymphoma related to modifiable lifestyle factors (primarily physical activity and diet).

Major Concerns:

My primary concern with this review is that a significant portion of the available literature for review may have been missed through the strict definition of non-Hodgkin lymphoma survivors to only include survivors of Diffuse Large B-cell lymphoma. This requirement forced the authors to eliminate the majority of potentially eligible articles related to NHL because they included other subtypes in addition to DLBCL. I believe the motivation to do so was the authors desire to have a uniform population representing the most common NHL subtype receiving intensive therapy; however, I believe including articles related to NHL survivors overall or limiting them to NHL survivors by treatment intensity (such as those treated with chemotherapy and/or radiation) may have accomplished a similar goal while allowing inclusion of additional literature focused on NHL.

Given the methods were already developed and described and the NHL population was limited to those with DLBCL, it would be helpful to better justify this decision using additional background information regarding the proportion of survivors who were treated for DLBCL and how they differ from survivors of other subtypes of NHL.

Similarly, if only studies with DLBCL survivors of NHL were included, why was the same criteria not applied to studies of ASCT? Where survivors of other subtypes of lymphoma who received ASCT in addition to cHL and NHL included? If so, I do not disagree with this inclusion but feel it is discordant to the reasoning used to exclude studies of NHL that were focused solely on DLBCL.

Minor Concerns:

1) I may be confused but cannot find the 10 included studies. It appears section 3.1 has 5 studies, 3.2 has 0 studies, 3.3 has 2 studies however 1 was previously included in section 3.1, 3.4 has 0 studies, 3.5 has 2 studies but both were included in section 3.1, and 3.6 has 1 study. This would be a total of 7 studies.

2) Introduction, Paragraph 3, Lines 82-88: Please correct populations discussed regarding reference 21 and 22. For reference 20, should specify statement "also of developing cardiovascular diseases" refers to cHL survivors as that was the population studied in reference 20. For references 21 and 22, these refer to survivors of Hodgkin lymphoma (not NHL as stated in the introduction).

3) Results, 3.1, paragraph 3 related to reference 20. The authors should consider the potential impact of survival bias on the results of this study, the survivors were diagnosed in 1965-1995 and included based on mailed questionnaire responses, it is very likely that a number of survivors (particularly from the more historic treatment eras) had a fatal 1st major cardiac event or another cause of death. Where the "matched controls" similar to the population with the cardiac outcome of interest in terms of the proportion who received radiation therapy or anthracycline chemotherapy? Where these treatment differences controlled for in the study?

4) Results, 3.5, paragraph 3. I do not follow this sentence, please rephrase. Perhaps, "No studies specifically address the issue of chronic fatigue among the survivors of DLBCL, although this assessment of this outcome may be limited as patients tend to be diagnosed with DLBCL at an older age." If that is the intended conclusion the authors wished to draw. 

5) Results, 3.6, paragraph 2. I suggest including some information regarding if the nurse-led survivorship intervention included other factors that may have lead to improved physical activity (not just generation of the SCP). There is conflicting literature on the utility of SCPs in terms of behavior change, improved screening etc and there may be a component of increased contact with health providers or in-person counseling provided by the nurses that also contributed to the improved outcomes, depending on the study methods. 

Author Response

Response to Reviewer  2

The authors performed a systematic review and nicely utilized PICOS (patient, intervention, control, outcome, study design) statements to specify the questions being addressed and followed PRISMA guidelines. The content of the review has potential clinical significance for providers of survivorship care and researchers focused on outcomes among adult survivors of adult Hodgkin lymphoma, and less so Diffuse Large B-cell Lymphoma related to modifiable lifestyle factors (primarily physical activity and diet).

The Authors thank the reviewer for the positive comment.

Major Concerns:

Question 1

My primary concern with this review is that a significant portion of the available literature for review may have been missed through the strict definition of non-Hodgkin lymphoma survivors to only include survivors of Diffuse Large B-cell lymphoma. This requirement forced the authors to eliminate the majority of potentially eligible articles related to NHL because they included other subtypes in addition to DLBCL. I believe the motivation to do so was the authors desire to have a uniform population representing the most common NHL subtype receiving intensive therapy; however, I believe including articles related to NHL survivors overall or limiting them to NHL survivors by treatment intensity (such as those treated with chemotherapy and/or radiation) may have accomplished a similar goal while allowing inclusion of additional literature focused on NHL.

Response to question 1

The Authors confirm that the choice of including data on DLBCL survivors and not on all the NHL is related to the clinical need of presenting evidence on a homogeneous cohort. These patients receive a homogeneous front-line treatment in the majority of cases, thus late onset toxicities can be well studies. In the meanwhile, DLBCL subtype accounts for about the 40% of all diagnosed NHL and have reached a 5-yrs OS of about 60%. Since the beginning of the study, this choice was largely discussed and shared among the multi-disciplinary group and the medological group who supported the research .

The Authors wanted also to specify that the other histotypes of NHL results more rare and not always they are considered curable like DLBCL (e.g. Mantle Cell Lymphoma).

The reasons of the choice have now been better presented in the Introduction at line 51. 

"In the context of haematological neoplasms, lymphomas represent the population that achieves the greatest survivor free from therapy, thanks to the constant amelioration of diagnostic and therapeutic strategies [3-5]. Classical Hodgkin lymphoma (cHL) usually manifests in the second-third decade of life, with an incidence of 2.3-2.6 new cases/100,000/year (esmo HL, sser). The disease in now curable in at least 80% of patients, with a 5-years overall survival (OS) of 88% (ESMO-NNC hl, seer). It represents one of the cancers for which a remarkable improve-ment has been observed in the last 40 years (nccn). Long-term survivors to cHL have been generally treated with conventional chemotherapy and radiotherapy, while data on long term toxicity of new targeted drugs will be known in a short time (ESMO, nccn). Diffuse Large B-cell lymphoma (DLBCL) is the most common lymphoid neoplasm in the adult and accounts for about the 40% of all non-Hodgkin's lymphoma (NHL). Its inci-dence ranges from is 3.8-5.6/100,000/year, with a median age at diagnosis of 70 years (ref seer). The 5-years OS actually reaches 60-64% (ref sito brit, seer). Survivors to DLBCL have been treated in the majority of cases with a standard induction chemotherapy based on CHOP or R-CHOP regimen according to the historical period. Salvage chemo-therapy and consolidation with autologous hematopoietic stem cell transplant (ASCT) represent the second-line treatment of eligible patients both for cHL and DLBCL [ref esmo DLBCL, ref NCCN DLBCL, esmo]. These two cohorts then represent the prevalent population of long-term lymphoma survivors [3] and constitute the population to which the systematic review we present has been addressed."  

And at line 108:

"The reasons for the inclusion of these two populations in the systematic review, largely discussed within the multi-disciplinary group, reside in the epidemiology, the cure rates and the possibility of analyzing cohorts of homogeneous and homogeneously treated patients, for whom long-term toxicities result better evaluable."

Question 2

Given the methods were already developed and described and the NHL population was limited to those with DLBCL, it would be helpful to better justify this decision using additional background information regarding the proportion of survivors who were treated for DLBCL and how they differ from survivors of other subtypes of NHL.

Response to question 2

The Authors thank for the comment. In the new version of the introduction, we have motivate the reasons for the choice to include only the DLBCL histotype among all NHL survivors and the related epidemiology.

"The reasons for the inclusion of these two populations in the systematic review, largely discussed within the multi-disciplinary group, reside in the epidemiology, the cure rates and the possibility of analyzing cohorts of homogeneous and homogeneously treated patients, for whom long-term toxicities result better evaluable."

Question 3

Similarly, if only studies with DLBCL survivors of NHL were included, why was the same criteria not applied to studies of ASCT? Where survivors of other subtypes of lymphoma who received ASCT in addition to cHL and NHL included? If so, I do not disagree with this inclusion but feel it is discordant to the reasoning used to exclude studies of NHL that were focused solely on DLBCL.

Response to question 3

The working group included in the initial search conducted with the Cochrane methodology also patients treated with ASCT as part of the second line treatment. The unique study including long-term survivors treated with ASCT and selected for the data extraction was by Stenehjem et al (2016) (ref 33). It was selected to analyse the effect of physical activity in long-term ASCT survors. 

In order to underlyne to limitations due to the inclusion of NHL and not only BLBCL, the Authors have added the following sentence before the risk of bias paragraph (line 248):

"It should be noted that the study lacked detailed reporting on histotypes of NHL, and therefore did not allow us to assess which outcomes referred to DLBCL. However, the study met the inclusion criteria, was the unique available in the subset of ASCT and provided markable conclusions in reference to specific chemotherapic agents, e.g. doxorubicin. "

Minor Concerns:

Question 4

1) I may be confused but cannot find the 10 included studies. It appears section 3.1 has 5 studies, 3.2 has 0 studies, 3.3 has 2 studies however 1 was previously included in section 3.1, 3.4 has 0 studies, 3.5 has 2 studies but both were included in section 3.1, and 3.6 has 1 study. This would be a total of 7 studies.

Response to question 4

We have corrected that overall the included study are 7, with some of them evaluable for multiple PICOs.

"Seven studies were ultimately included in this systematic review, some of them were eligible for multiple PICOs."

Question 5

2) Introduction, Paragraph 3, Lines 82-88: Please correct populations discussed regarding reference 21 and 22. For reference 20, should specify statement "also of developing cardiovascular diseases" refers to cHL survivors as that was the population studied in reference 20. For references 21 and 22, these refer to survivors of Hodgkin lymphoma (not NHL as stated in the introduction).

Response to question 5

The Authors have modified the sentence as follows:

Line 81:

"Moreover, a large population study conducted on 3,970 B-NHL shows that a normal body weight is associated with a reduced risk of developing diabetes mellitus [19]. Obesitiy represented a risk factor for cardiovascular diseases in a cohort of 2,617 5-years cHL survivors [20]."

Line 85:

" Chronic fatigue has been documented in about 30% of cHL survivors and is associated with early cognitive impairment [21,22]."

Question 6

3) Results, 3.1, paragraph 3 related to reference 20. The authors should consider the potential impact of survival bias on the results of this study, the survivors were diagnosed in 1965-1995 and included based on mailed questionnaire responses, it is very likely that a number of survivors (particularly from the more historic treatment eras) had a fatal 1st major cardiac event or another cause of death. Where the "matched controls" similar to the population with the cardiac outcome of interest in terms of the proportion who received radiation therapy or anthracycline chemotherapy? Where these treatment differences controlled for in the study?

Response to question 6

According the reference 20, study by van Nimwegen et al 'Radiation Dose-Response Relationship for Risk of Coronary Heart' J Clin Oncol 34:235-243.2016, 

The authors declared in the manuscript that:

"We conducted a nested case-control study in a cohort of 2,617 5-year HL survivors, treated between 1965 and 1995. ..."

"Cases were patients diagnosed with CHD as their first cardiovascular event after HL."

Cases (n=325) were patients who developed CHD in the form of either symptomatic myocardial infarction or angina pectoris requiring intervention (Common Terminology Criteria for Adverse Events, version 4.0; Appendix Text A1, online only) as their first clinically significant heart disease. Cases were identified from medical records or postal questionnaires completed by their general practitioners. Follow-up was complete up to October 2013.

From Statistical Analysis:

Odds ratios for CHD for different levels of each factor were calculated using conditional logistic regression on sets of individual cases and their matched controls, and were interpreted as rate ratios (RRs). The Wald method was used to calculate 95% CIs for factors with two levels. The amount of information in each category, including the reference category (so-called floating absolute risks), was used to calculate 95% CIs for factors with more than two levels. Multivariable regression was used to assess and control for confounding and to evaluate interactions between radiation dose and other factors.

Control were patients free of cardiac diseases.

"For each case with CHD, we attempted to select four controls from the cohort, individually matched on sex, age atHL diagnosis (# 1 year), and date of HL diagnosis (# 3 years). Controls had to be free of any cardiac disease grade 2 at the cutoff date. In total, 1,204 controls were matched to the cases."

Treatments were variable (e.g. radiotherpay, chemotherapy, alkylating agents, Anthracyclines and other) controlled for both case and control with multivariate regression analysis. ["Multivariable regression was used to assess and control for confounding and to evaluate interactions between radiation dose and other factors."]

The paragraph (line 194) has then been modified as follow:

"One nested case-control study presented the cardiovascular late toxicity outcome and discussed the impact of lifestyle factors [20] in a large cohort of 2,617 5-year cHL survivors treated between 1965 and 1995. Follow-up was complete up to October, 2013. The study evaluated the risk of coronary heart disease (CHD) as first cardiovascular event after lymphoma, according to the radiation dose to the heart and type of chemotherapy. An estimate of the in-field irradiated heart volume was calculated via radiation charts and simulation radiographs. Other clinical and lifestyle factors were collected from medical records and from mailed questionnaires completed by their general practitioners (70% of responses). From the whole cohort, 325 survivors reported a CHD as first event (median interval from lymphoma 19 years). For each patient with CHD, four controls, who had not developed cardiac disease and matched for sex, age and date of HL diagnosis, had been selectd (n=1,204).Treatments were variable but controlled for both case and control with multivariate regression analysis."

Question 7

4) Results, 3.5, paragraph 3. I do not follow this sentence, please rephrase. Perhaps, "No studies specifically address the issue of chronic fatigue among the survivors of DLBCL, although this assessment of this outcome may be limited as patients tend to be diagnosed with DLBCL at an older age." If that is the intended conclusion the authors wished to draw.

Response to question 7

The Authors have rephrased as follow:

"No studies specifically address the issue of preventive measures on chronic fatigue among DLBCL survivors. "

Question 8

5) Results, 3.6, paragraph 2. I suggest including some information regarding if the nurse-led survivorship intervention included other factors that may have lead to improved physical activity (not just generation of the SCP). There is conflicting literature on the utility of SCPs in terms of behavior change, improved screening etc and there may be a component of increased contact with health providers or in-person counseling provided by the nurses that also contributed to the improved outcomes, depending on the study methods.

Response to question 8

The authors considered this aspect suggested by the reviewer and modified the text:

"The main outcome of the analysis was to enhance the awareness and adoption of healthy lifestyles in a population of cHL survivors through a nurse-led survivorship intervention, including the development and delivery of a tailored SCP and evaluated by the General Health Index and Health Promoting Lifestyle Profile II at 4 time points. The nurse-led consultations also included an education package tailored to individuals' health needs. This study was published in 2013 and recruited 30 cHL survivors and 30 healthy controls. In response to the defined PICOs, from baseline to 6 months post-intervention, a statistically significant benefit was detected for some domains: improvement of physical activity (P=0.014), nutrition (P=0.0005), health promoting lifestyle (P=0.005). The intervention was feasible and demonstrated significant potential to improve awareness of health status and healthy lifestyle behaviors in this subset of patients [36]. It cannot be concluded that the benefits presented by the patients were due only to the administration of the SCP but that the consultations and the educational package may have played a synchronous role."

Reviewer 3 Report

Authors performed a systematic literature review to assess evidence about the usefulness of healthy lifestyle to reduce the impact of late sequelae among lymphoma (cHL and DLBCL) survivors.  Only 10 studies were included in the final review since several articles were excluded for different reasons while for other research questions no studies were considered evaluable for analysis.  Overall conclusion was that regular physical activity has shown to have a protective effect in reducing cardiovascular complications, while being overweight was associated to increased risk of coronary artery disease.  No data were available about the effect of an healthy (Mediterranean) diet.  Results are quite expected and show the lack of specific exhaustive data regarding lymphoma survivors.

While the methods and results sections seems well structured and organized, the introduction part seems a bit vague and might need a further English editing and organization

Overall the manuscript is informative

Specific comments:

Abstract:

lines 29 and 31: authors refer to the “correction” of lifestyles assuming that all survivors have a unhealthy lifestyle.  The word “promotion” seems more adequate.

Line 36: autologous stem cell transplant (ASCT) suddenly “pops-up” in the results section but no mention was made before about it, reader would expect some results regarding DLBCL.  Moreover, ASCT is a type of treatment, not a disease, and indeed the Medline review was based on these keywords.  I suggest to rephrase the results section focusing on the two types of lymphoma considered.

Introduction:

Line 50: the estimate of 16M survivors refers to the US population, this should be clarified.

Overall this section does not flow smoothly and some sentences are a bit e.g. line 79-80: the sentence of physical activity ameliorating QoL in lymphoma survivors seems out of context in the paragraph. I suggest to delete or move after the first sentence of the paragraph (line 73-74).

Lines 108-109: the reason of including in this study only cHL and DLBCL is very briefly summarized in  a short sentence.  I think a bit more time should be given to explain to the reader the reason of this decision.

Materials and methods:

Lines 171-74: Table 1 correctly reports the PICOs of each clinical question. I think it should be referred to Table 1 in the second sentence of the paragraph.  My suggestion would be: “For each clinical question, the pertinent PICO was defined  (Table 1) and the included studies …. “

Results:

Incorporating evidence tables (as online material) for each manuscript included in the review would help the reader.  Overall for each question it would be appropriate to provide the level of evidence decided by the group as well the grade (from A: strong in favor to E strong against” of recommendation

Lines 277-287: several details are given about a study which was an abstract only publisher already several years ago to which was apparently not followed by a peer reviewed manuscript.  I think that those data should be used with caution

Line 313: refer to “chapter 3.1” or use the same wording “

Lines 335-347: as before, these results are based on an abstract and authors correctly conclude that quality is low.  I do not think that any recommendation should be based on these results.

Discussion:

Line 358 (and also 396): authors conclude that the review “allowed to evaluate the efficacy of adherence to healthy lifestyles” … and then state (correctly) that the Mediterranean diet or balanced nutrition are among the modifiable factors that might help in tertiary prevention.  From the text in the paragraph it might seem that the authors have found evidence of the positive effect of the diet in the prevention of late sequelae in lymphoma survivors.  This (unfortunately) is not the case because of lack of data, but this should be mentioned since from the data no recommendation may be given.

Author Response

Response to Reviewer 3

Authors performed a systematic literature review to assess evidence about the usefulness of healthy lifestyle to reduce the impact of late sequelae among lymphoma (cHL and DLBCL) survivors.  Only 10 studies were included in the final review since several articles were excluded for different reasons while for other research questions no studies were considered evaluable for analysis.  Overall conclusion was that regular physical activity has shown to have a protective effect in reducing cardiovascular complications, while being overweight was associated to increased risk of coronary artery disease.  No data were available about the effect of an healthy (Mediterranean) diet.  Results are quite expected and show the lack of specific exhaustive data regarding lymphoma survivors. 

While the methods and results sections seems well structured and organized, the introduction part seems a bit vague and might need a further English editing and organization

Overall the manuscript is informative

The Authors thank the Review for the positive comments.

We have extensively modified the introduction, we report here the first part:

“Survivorship is becoming an important issue of cancer care. It is estimated that the number of cancer survivors has now reached more than 16 million in US and 12 million in Europe and shows a constant increase over time [1-5]. The majority of survivors have been cured for breast, prostate, colorectal cancer and melanoma, accounting for about the 58% of survivors [2]. In the context of haematological neoplasms, lymphomas represent the population that achieves the greatest survivor free from therapy, thanks to the constant amelioration of diagnostic and therapeutic strategies [3-7].

Classical Hodgkin lymphoma (cHL) usually manifests in the second-third decade of life, with an incidence of 2.3-2.6 new cases/100,000/year [8-9]. The disease in now curable in at least 80% of patients, with a 5-years overall survival (OS) of 88% [8-10]. It represents one of the cancers for which a remarkable improvement has been observed in the last 40 years [10]. Long-term survivors to cHL have been generally treated with conventional chemotherapy and radiotherapy, while data on long term toxicity of new targeted drugs will be known in a short time [8,10]. Diffuse Large B-cell lymphoma (DLBCL) is the most common lymphoid neoplasm in the adult and accounts for about the 40% of all non-Hodgkin's lymphoma (NHL). Its incidence ranges from is 3.8-5.6/100,000/year, with a median age at diagnosis of 70 years [11]. The 5-years OS actually reaches 60-64% [3-11]. Survivors to DLBCL have been treated in the majority of cases with a standard induction chemotherapy based on CHOP (cyclophosphamide, vincristine, doxorubicine, prednisone) or rituximab-CHOP regimen according to the historical period. Salvage chemotherapy and consolidation with autologous hematopoietic stem cell transplant (ASCT) represent the second-line treatment of eligible patients both for cHL and DLBCL [12-13]. These two cohorts then represent the prevalent population of long-term lymphoma survivors [3] and constitute the population to which the systematic review we present has been addressed.

Long-term lymphoma survivors could develop a series of late sequelae, mainly represented by cardiovascular, endocrine-metabolic and neurological toxicities, secondary cancers and infertility [14,15]. The genesis of these late toxicities is multi-factorial and could recognize two main groups of risk factors: i) not modifiable factors: chemotherapy, radiation therapy and ASCT; family history; age [16,17]; and ii) modifiable factors, such as unhealthy lifestyle factors [18,19].

Lifestyles represent an emerging topic both in cancer prevention and in the prevention of late sequelae to chemo and radiotherapy. In this context, it is essential to identify the unhealthy lifestyles of long-term survivors in order to ameliorate educational and preventive measures, reduce the risk of subsequent diseases, and improve quality of life (QoL). Most of the published experiences in this sense concern patients treated from solid tumors, in particular breast and colorectal cancer. For some cancer types, a healthy lifestyle has been associated with a reduced risk of recurrence and mortality [2]. In the subset of cancer survivorship, individual lifestyle behaviors, such as normal body weight, physical activity, smoking, diet quality, have been associated with reduced mortality [20,21]. The meta-analysis by Mishra et coll. demonstrated a beneficial effect of exercise on Health Related Quality of Life (HRQoL) in 40 trials including heterogeneous cohorts of cancer survivors [22].”  “...”

The text has been reviewed by a native English speaker.

Specific comments:

Abstract:

Question 1

lines 29 and 31: authors refer to the “correction” of lifestyles assuming that all survivors have a unhealthy lifestyle.  The word “promotion” seems more adequate.

Response to question 1

The Authors have now substituted the word "correction" with "promotion" in the abstract section. All the changes can be visible in blue font.

Question 2

Line 36: autologous stem cell transplant (ASCT) suddenly “pops-up” in the results section but no mention was made before about it, reader would expect some results regarding DLBCL.  Moreover, ASCT is a type of treatment, not a disease, and indeed the Medline review was based on these keywords.  I suggest to rephrase the results section focusing on the two types of lymphoma considered.

Response to question 1

Line 33: in order to introduce the aspect of ASCT, the Authors have now included at line 33 the sentence:

"treated at adult age with first line or second line therapy including autologous stem cell transplant (ASCT)."

Line 34: the sentence has been rephrased as follows:

"Ten studies were ultimately included in this systematic review. The majority of the studies emerged from data extraction regarding cHL, while less evidence resulted for the DLBCL survivors. Five studies in favour of physical activity provided consistent data for a reduction of the cardiovascular risk in cHL and also in survivors who underwent ASCT.A beneficial effect of physical activity in reducing chronic fatigue was found."

Introduction:

Question 3

Line 50: the estimate of 16M survivors refers to the US population, this should be clarified.

Response to question 3

Thank you for the correction. The Authors have modified the sentence as follows:

"..16 million in US and 12 million in Europe"

A new reference reporting the updated numbers of cancer survivors in Europe has been added:

https://ec.europa.eu/info/strategy/priorities-2019-2024/promoting-our-european-way-life/european-health-union/cancer-plan-europe_en

Question 4

Overall this section does not flow smoothly and some sentences are a bit e.g. line 79-80: the sentence of physical activity ameliorating QoL in lymphoma survivors seems out of context in the paragraph. I suggest to delete or move after the first sentence of the paragraph (line 73-74).

Response to question 4

To make the sentence more clear, it has been modified from line 79:

"Some unhealthy lifestyles seam to affect different aspects of the patient's life.It has been reported thata regular physical activity ameliorates QoL and psychological and physical fitness in classical Hodgkin lymphoma (cHL) and non-Hodgkin’s lymphoma (NHL) survivors [15,16,18].Moreover, a large population study conducted on 3,970 B-NHL shows that a normal body weight is associated with a reduced risk of developing diabetes mellitus [19]. Obesitiy represented a risk factor for cardiovascular diseases in a cohort of 2,617 5-years cHL survivors [20]."

Question 5

Lines 108-109: the reason of including in this study only cHL and DLBCL is very briefly summarized in  a short sentence.  I think a bit more time should be given to explain to the reader the reason of this decision.

Response to question 5

We thank the reviewer for this comment. Now we have motivated the reason for choising cHL and DLBCL in the systematic review:

Introduction, line 51. 

"In the context of haematological neoplasms, lymphomas represent the population that achieves the greatest survivor free from therapy, thanks to the constant amelioration of diagnostic and therapeutic strategies [3-5]. Classical Hodgkin lymphoma (cHL) usually manifests in the second-third decade of life, with an incidence of 2.3-2.6 new cases/100,000/year (esmo HL, sser). The disease in now curable in at least 80% of patients, with a 5-years overall survival (OS) of 88% (ESMO-NNC hl, seer). It represents one of the cancers for which a remarkable improve-ment has been observed in the last 40 years (nccn). Long-term survivors to cHL have been generally treated with conventional chemotherapy and radiotherapy, while data on long term toxicity of new targeted drugs will be known in a short time (ESMO, nccn). Diffuse Large B-cell lymphoma (DLBCL) is the most common lymphoid neoplasm in the adult and accounts for about the 40% of all non-Hodgkin's lymphoma (NHL). Its inci-dence ranges from is 3.8-5.6/100,000/year, with a median age at diagnosis of 70 years (ref seer). The 5-years OS actually reaches 60-64% (ref sito brit, seer). Survivors to DLBCL have been treated in the majority of cases with a standard induction chemotherapy ba-sed on CHOP or R-CHOP regimen according to the historical period. Salvage chemo-therapy and consolidation with autologous hematopoietic stem cell transplant (ASCT) represent the second-line treatment of eligible patients both for cHL and DLBCL [ref esmo DLBCL, ref NCCN DLBCL, esmo]. These two cohorts then represent the prevalent population of long-term lymphoma survivors [3] and constitute the population to which the systematic review we present has been addressed."  

Line 108:

"The reasons for the inclusion of these two populations in the systematic review, largely discussed within the multi-disciplinary group, reside in the epidemiology, the cure rates and the possibility of analyzing cohorts of homogeneous and homogeneously treated patients, for whom long-term toxicities result better evaluable."

Materials and methods:

Question 6

Lines 171-74: Table 1 correctly reports the PICOs of each clinical question. I think it should be referred to Table 1 in the second sentence of the paragraph.  My suggestion would be: “For each clinical question, the pertinent PICO was defined  (Table 1) and the included studies …. “

Response to question 6

Thank you for the advise. We have modofied the sentence as follows (from line 171):

"For each clinical question, the pertinent PICO was defined (Table 1) and the included studies providing relevant information were summarized narratively and tabulated to highlight similarities and differences in methods and results."

Results:

Question 7

Incorporating evidence tables (as online material) for each manuscript included in the review would help the reader. 

Response to question 7

Thanks for your suggestion but to help the redear, we prefer to describe results of evidence table in the full manuscript and to leave evidence table as supplementary, considering the big amount of data about the studies included in this review and the outcomes extracted.

Question 8

Overall for each question it would be appropriate to provide the level of evidence decided by the group as well the grade (from A: strong in favor to E strong against” of recommendation

Response to question 8

We did not conduct a formal guideline and our aim was not to make recommendations with GRADE or other EBM approach. Due to the lack of systematic reviews to summarise the evidence in this field, we decide to start with this step and probably guideline will be our next project.

Anyway we include all formal step according Cochrane methods to conduct the systematic review including assessment of risk of bias.

Question 9

Lines 277-287: several details are given about a study which was an abstract only publisher already several years ago to which was apparently not followed by a peer reviewed manuscript.  I think that those data should be used with caution

Response to question 9

In order to underline this aspect, the Authors have added at line 188 the sentence:

"It must be reiterated that the results have been reported only through an abstract, whose full data have not yet been published."

Question 10

Line 313: refer to “chapter 3.1” or use the same wording “

Response to question 10

"Data on chronic fatigue have partially been exposed previously in Chapter 3.1."

Question 11

Lines 335-347: as before, these results are based on an abstract and authors correctly conclude that quality is low.  I do not think that any recommendation should be based on these results.

Response to question 11

No recommendations were drawn from these results.

We then modified the sentence at row 332-333 as follows:

"The included study was a prospective cohort study, published as an abstract with relevant limitations."

Authors also included the following sentence in the discussion line 412:

"According to the literature, data supporting the benefit of the use of SCP in lymphoma survivors are still insufficiet and there is a need for future evaluations."

Discussion:

Question 12

Line 358 (and also 396): authors conclude that the review “allowed to evaluate the efficacy of adherence to healthy lifestyles” … and then state (correctly) that the Mediterranean diet or balanced nutrition are among the modifiable factors that might help in tertiary prevention.  From the text in the paragraph it might seem that the authors have found evidence of the positive effect of the diet in the prevention of late sequelae in lymphoma survivors.  This (unfortunately) is not the case because of lack of data, but this should be mentioned since from the data no recommendation may be given.

Response to question 12

From line 393, the sentence has been largely modified, in order to underline that even if there are no evidence emerging from the sistematic review due to the lack of eligible studies, it is rational to advice a balance diet:

"Compared to solid tumors, well-designed studies aiming to evaluate the impact of the Mediterranean diet on late toxicities and secondary cancers in cHL and DLBCL long-term survivors are lacking. In any case, general guidelines recommend adhesion to this alimentary regimen for all cancer survivors [2,13,37,38]. Even if from our systematic review no studies that have as their topic the benefit of the Mediterranean diet have emerged, it is rational to advise a controlled and balanced diet, consistent with the Mediterranean diet, to lymphoma survivors."

Round 2

Reviewer 3 Report

I'm satisfied with the revised manuscript in which all my comments have been adequately answered and/or incorporated in the text

I think that the manuscript can be accepted for publication in Cancers .